# Effect of Geographic Regions on the Flavor Quality and Non-Volatile Compounds of Chinese Matcha

**DOI:** 10.3390/foods14010097

**Published:** 2025-01-02

**Authors:** Hongchun Cui, Yun Zhao, Hongli Li, Min Ye, Jizhong Yu, Jianyong Zhang

**Affiliations:** 1Tea Research Institute, Hangzhou Academy of Agricultural Science, Hangzhou 310024, China; chc1134@126.com (H.C.);; 2Tea Research Institute, Chinese Academy of Agricultural Science, Hangzhou 310008, China

**Keywords:** matcha, regions, sensory quality, flavor, non-volatile components, characteristic

## Abstract

Matcha is a very popular tea food around the world, being widely used in the food, beverage, health food, and cosmetic industries, among others. At present, matcha shade covering methods, matcha superfine powder processing technology, and digital evaluations of matcha flavor quality are receiving research attention. However, research on the differences in flavor and quality characteristics of matcha from the same tea tree variety from different typical regions in China is relatively weak and urgently required. Taking Japan Shizuoka matcha (R) as a reference, the differences in sensory quality characteristics and non-volatile substances of matcha processed with the same tea variety from different regions in China were analyzed. The samples were China Hangzhou matcha (Z1), China Wuyi matcha (Z2), China Enshi matcha (H), and China Tongren matcha (G), which represent the typical matcha of eastern, central, and western China. A total of 1131 differential metabolites were identified in the matcha samples, comprising 118 flavonoids, 14 tannins, 365 organic acids, 42 phenolic acids, 22 alkaloids, 39 saccharides, 208 amino acids and derivatives, 17 lignans and coumarins, seven quinones, 44 nucleotides and derivatives, 14 glycerophospholipids, two glycolipids, 15 alcohols and amines, 140 benzenes and substituted derivatives, 38 terpenoids, 30 heterocyclic compounds, and 15 lipids. Kaempferol-7-O-rhamnoside, 3,7-Di-O-methylquercetin, epigallocatechin gallate, epicatechin gallate, and epigallocatechin were detected in Z1, Z2, H, and G. A total of 1243 metabolites differed among Z1, Z2, and R. A total of 1617 metabolites differed among G, H, and R. The content of non-volatile difference metabolites of Z2 was higher than that of Z1. The content of non-volatile difference metabolites of G was higher than that of H. The 20 key differential non-volatile metabolites of Z1, Z2, G, and H were screened out separately. The types of non-volatile flavor differential metabolites of G and H were more numerous than those of Z1 and Z2. The metabolic pathways, biosynthesis of secondary metabolites, biosynthesis of co-factors, flavonoid biosynthesis, biosynthesis of amino acids, biosynthesis of various plant secondary metabolites, and purine metabolism of metabolic pathways were the main KEGG pathways. This study provides new insights into the differences in metabolite profiles among typical Chinese matcha geographic regions with the same tea variety.

## 1. Introduction

China is the largest producer of matcha and the largest consumer of matcha products in the world, surpassing even Japan, which is the second largest producer. There are significant differences between matcha from China and that from other countries in terms of the production process, choice of ingredients, and flavor. Matcha not only has the unique characteristics of a bright color, rich taste, and fragrant smell [1] but also has a variety of biological activities such as anti-anxiety [2,3], stress reduction [4,5], anti-cancer [6,7], hypoglycemic [8,9], antioxidant [9], and anti-bacterial activities [10], as well as the ability to treat and prevent infectious diseases [11].

The sensory quality of matcha has the characteristics of “three clear”, namely, fragrance, clear mouth, and slightly green (grass) gas. Matcha is made from small tea leaves that have not been kneaded [12]. Covering and steaming are two characteristic processing steps in matcha processing, and they are also key processes in the formation of matcha flavor and quality [13]. A trellis must be set up 20 days before picking tea in spring. The shading rate based on covering reed curtains and straw curtains is over 98% [14,15]. The content of non-volatile components of unshaded tea is significantly lower than that of shaded tea, and these components include chlorophyll, amino acids, total amino, and chlorophyll [15]. Matcha provides a full intake of tea nutrition and health components. Matcha maximizes the retention of tea polyphenols, caffeine, free amino acids, tea polysaccharides, chlorophyll, aromatic substances, fiber, protein, and a large number of minerals, vitamins, and other biologically active substances, including water-soluble and non-water-soluble substances.

Compared with unshaded tea products, matcha contains fewer bitter catechins and more amino acid components [16,17,18]. The typical flavor characteristics of matcha are a sweet taste, freshness, tenderness, and astringency [19,20]. Water-soluble carbohydrates, organic acids, amino acids, tea polyphenols, flavonol glycosides, tannins, tannic acid and flavonoid glycosides, and caffeine are the main contributors to the taste quality of matcha [21,22,23,24]. The main chemical components that contribute to the bitter and astringent flavor of matcha are catechins [25,26,27]. The most active and abundant compound of catechins is EGCG, followed by ECG and EC [28,29]. Catechins are categorized into esters and non-esters based on their structure. The ester catechins are the most important bitter and astringent substances in matcha [30]. The non-ester catechins are more mellow and less astringent [31,32].

The refreshing taste of matcha is mainly due to the high concentration of free amino acids [33,34,35]. The main amino acids in matcha are theanine, glutamic acid, aspartic acid, and threonine. The aspartic acid are the main amino acids that contribute to its umami taste. The synergistic effect of glutamic acid and aspartate with threonine contributes to the quality of matcha [35,36,37]. The umami-enhancing compounds of matcha include theanine, succinic acid, gallic acid, and theogallin, which increase the umami intensity of glutamate in proportion. Sweet amino acids and carbohydrates are the main contributors to the sweetness of matcha. However, the concentration of sweet compounds in matcha soup is very low, and the dose above threshold value (Dot value) is lower than 0.1 [33].

Matcha is often used as a food additive or food ingredient and is applied in beverages, biscuits, breads, and other foods. Traditional matcha has been processed using a stone mill since the early 19th century, and its particle size is 106 μm [37]. In recent years, ultrafine grinding technology has been used to process matcha with a particle size of 10–32 μm [38]. Matcha can be suspended in hot water without settling. Even after a long storage period, matcha still does not produce precipitation. It has a significant characteristic of being ultrafine, with high levels of nutritive health substances.

To detect the chemical components in matcha and evaluate its quality, high-performance liquid chromatography [37,38,39], hyperspectral microscope imaging technology [36,40], and near-infrared spectroscopy [41] have been used, mainly to analyze a relatively small number of flavor components of matcha [37,38,39]. Current matcha studies largely focus on shading technology, ultrafine grinding technology, and applications in the food and cosmetics industries [42,43,44,45]. The understanding of the quality characteristics of matcha processed from the same variety from different geographic regions in China is still incomplete. In this study, typical matcha samples were processed using the same processing techniques employed in the four main production areas in China, namely, Hangzhou, Wuyi, Enshi, and Tongren. The varieties of tea trees used in the four regions were the same. The effects of these geographic regions on the sensory characteristics and non-volatile substances of matcha were systematically analyzed. We hope to provide new insights into the complex chemical substance base of matcha and a theoretical basis for research on the formation and regulation of matcha quality.

## 2. Materials and Methods

### 2.1. Materials

Methanol and acetonitrile (chromatographic pure grade) were purchased from Thermo Fisher Scientific Inc. (Waltham, MA, USA). Formic acid, 2-amino-3-(2-chloro-phenyl)-propionic acid, and ammonium formate were purchased from Sigma Aldrich Ltd. (Shanghai, China).

### 2.2. Instruments

An Ultimate 3000 UHPLC System and a Q Exactive system were purchased from Thermo Fisher Scientific UK Ltd. (Waltham, MA, USA). A tissue pulverizer was purchased from Wenling Linda Machinery Co. Ltd. (Taizhou, China). An ultrasonic cleaner was supplied by Shanghai Kedao Ultrasonic Instrument Co. Ltd. (Shanghai, China). Centrifuges were purchased from Eppendorf Ltd. (Hamburg, Germany). A vortex mixer was purchased from Thermo Fisher Scientific Inc. (Waltham, MA, USA).

### 2.3. Matcha Processing Methods in Different Regions

The fresh tea leaves of shaded tea trees were harvested from tea gardens located in Tongren city, Enshi city, Hangzhou city, and Wuyi city in China. The tea tree varieties of the abovementioned fresh leaves in the 4 geographic regions were all Fudingdabai tea. The matcha processing technology employed was the same for each of the 4 geographic regions. The matcha processing technology was “shaded tea fresh leaves → steaming green → grinding → ultrafine grinding → low-temperature drying → matcha”. The matcha products processed in the 4 geographic regions were named Tongren matcha (G), Enshi matcha (H), Hangzhou matcha (Z1), and Wuyi matcha (Z2). Japan Shizuoka matcha (R) was the control sample. It was made from the fresh leaves of Yabukita tea tree varieties using the same processing technology as described above.

### 2.4. Sensory Evaluation of Matcha Sample

First, 0.6 g of the matcha samples was separately weighed and placed in a 240 mL tea bowl. Then, 150 mL of boiling water was poured into a tea bowl and kept for 3 min. The evaluation team consisted of four men and three women. They were all qualified volunteers with the ability to provide professional tea quality evaluation [9]. The password review method was used to evaluate the test results. The matcha liquor was stirred. Then, the liquor color, aroma, and taste of the matcha samples were evaluated by the 7 professional tea tasters in turn. The evaluation method of sensory quality was GB/T 23776-2018 [46] from China. Matcha was assigned 10% for appearance, 20% for liquor color, 35% for aroma, and 35% for taste.

The total sensory score of matcha samples was calculated according to Formula (1):(1)Totalsensoryscore=A×10%+B×20%+C×35%+D×35%

*A* represents the appearance score, *B* represents the liquor color score, *C* represents the aroma score, and *D* represents the taste score.

### 2.5. Sample Preparation

Matcha samples from the different geographic regions were accurately weighed and added to a 2 mL centrifuge tube. Then, 600 µL MeOH, 2-amino-3-(2-chloro-phenyl)-propionic acid (4 ppm), and 100 mg glass beads were added to the centrifuge tube and swirled for 30 s. The solution was processed in the vortex mixer at 60 Hz for 90 s and sonicated for 10 min at 25 °C. The sonicated solution was centrifuged at 12,000 rpm for 10 min at 4 °C. The centrifugation supernatant was filtered with a 0.22 μm membrane.

### 2.6. Liquid Chromatography Conditions

An ACQUITY UPLC^®^ HSS T3 (150 × 2.1 mm, 1.8 µm) (Milford, NH, USA) was used to analyze the non-volatile substances of the matcha. The column temperature was 40 °C. The flow rate was set at 0.25 mL/min. The mobile phase A was acetonitrile with 0.1% formic acid (*v*/*v*). The mobile phase B was ultrapure water with 0.1% formic acid (*v*/*v*). The linear elution gradient was the following: 1 min, 2% A; 9 min, 50% A; 12 min, 98% A; 13.5 min, 98% A; 14 min, 2% A; 17 min, 2% A [47].

### 2.7. Mass Spectrum Conditions

The spray voltage for the positive ionization mode ESI(+) was 3.50 kV. The spray voltage for the negative ionization mode ESI(−) was −2.50 kV [47]. The sheath gas flow rate was 35 arb. The auxiliary gas flow rate was 10 arb. The capillary temperature was 325 °C. The scanning mode, Full MS/dd-MS2 mode, was automatic. The scanning range was 100–1000 *m*/*z*. The Full MS resolution was 70,000. The MS/MS resolution was 17,500. The dynamic exclusion time modeling was automatic.

### 2.8. Data Processing

The experiments relating to the sensory evaluation scores and metabolomic data of the matcha samples were repeated 3 times. The results of the repeated experiments are expressed as the mean ± standard deviation. SPSS 25 software (IBM, NY, USA) was used for significance analysis. An ANOVA test was used for one-way analysis of variance. The least significant difference was performed at the significance level *p* < 0.01.

## 3. Results

### 3.1. Sensory Quality Analysis of Different Geographic Regions of Matcha

The sensory quality evaluation results of the matcha samples from different geographic regions are presented in Table 1. The total quality score, appearance score, liquor color score, taste score, and aroma score of H, G, Z1, and Z2 were lower than R, which is consistent with previous research [1]. The differences in the appearance score, liquor color score, taste score, and total score between the China Wuyi matcha (Z2) and China Enshi matcha (H) were not significant (*p* > 0.01) and were higher than the China Tongren matcha (G) and China Hangzhou matcha (Z1). The differences in the appearance score, liquor color score, taste score, and total score between the China Wuyi matcha (Z2) and China Enshi matcha (H) were significant (*p* < 0.01). The differences in the appearance score, liquor color score, taste score, and aroma score between the China Tongren matcha (G) and China Hangzhou matcha (Z1) were not significant (*p* > 0.01). The differences in the liquor color score, taste score, and total score between the China Tongren matcha (G) and China Hangzhou matcha (Z1) were not significant (*p* > 0.01). The differences in the appearance score and aroma score between the China Tongren matcha (G) and China Hangzhou matcha (Z1) were significant (*p* < 0.01).

The sensory quality evaluation comments on the matcha tea from different geographic regions were compared (Table 1). The appearance of the Japan Shizuoka matcha (R) and Chinese matcha (H, Z2, G, Z1) was delicate and even. The appearance of the Japan Shizuoka matcha (R) was characterized by a vibrant green color. It is worth mentioning that the appearance of the China Tongren matcha (G) was characterized by a tender green color. The liquor color profile of all the matcha samples was green. The liquor color profile of the China Wuyi matcha (Z2) and China Enshi matcha (H) was a more intense green than that of the China Tongren matcha (G) and China Hangzhou matcha (Z1). The matcha samples from the different geographic regions had the characteristic of a seaweed fragrance. The seaweed fragrance and freshness of the China Enshi matcha (H) were higher than the China Wuyi matcha (Z2), China Tongren matcha (G), and China Hangzhou matcha (Z1). Compared with the China Tongren matcha (G) and China Hangzhou matcha (Z1), the taste profile of the China Wuyi matcha (Z2) and China Enshi matcha (H) was not astringent.

Based on the above sensory quality evaluation score and comments, the China Enshi matcha (H) and China Wuyi matcha (Z2) had a better flavor than the China Hangzhou matcha (Z1) and China Enshi matcha (H), which mainly manifested in aroma, taste, and liquor color.

### 3.2. Cluster Analysis of Non-Volatile Substances of Matcha from Different Geographic Regions

Differences in the accumulation patterns of metabolites in matcha samples from different geographic regions were analyzed using clustering in a heat map (Figure 1). There were significant differences in the non-volatile metabolites in the matcha from different geographic regions, which were divided into a total of three clusters (Figure 1a). Cluster 1 was China Hangzhou matcha (Z1). Cluster 2 was China Wuyi matcha (Z2) and Japan Shizuoka matcha (R). Cluster 3 was China Enshi matcha (H) and China Tongren matcha (G). The non-volatile composition of the samples clustered into each group was consistent.

Figure 1b shows a heat map of the composition and content difference of the non-volatile substances in the matcha samples from different geographic regions. A total of 1131 distinct metabolites were identified in the samples, comprising 118 flavonoids, 14 tannins, 365 organic acids, 42 phenolic acids, 22 alkaloids, 39 saccharides, 208 amino acids and derivatives, 17 lignans and coumarins, seven quinones, 44 nucleotides and derivatives, 14 glycerophospholipids, two glycolipids, 15 alcohols and amines, 140 benzenes and substituted derivatives, 38 terpenoids, 30 heterocyclic compounds, and 15 lipids. Organic acids, amino acids and derivatives, benzene and substituted derivatives, flavonoids, and saccharides were the main non-volatile substances in the matcha samples. These five types of substances accounted for more than 76.9% of the non-volatile substances in the samples. The nucleotides and derivatives, phenolic acids, terpenoids, heterocyclic compounds, and alkaloids accounted for more than 15.6% of the non-volatile substances in the samples. The other 12 kinds of substances accounted for 7.5% of the non-volatile substances.

The metabolite percentages in the matcha samples are shown in Figure 2. Alcohols and amines made up 1.2%, alkaloids 1.76%, amino acids and derivatives 16.65%, benzene and substituted derivatives 11.13%, free fatty acids 0.16%, flavonoids 9.45%, glycolipids 0.16%, glycerophospholipids 1.12%, heterocyclic compounds 2.4%, lignans and coumarins 1.36%, lipids 1.2%, nucleotides and derivatives 3.52%, organic acids 29.14%, saccharides and ketone compounds 12.25%, phenolic acids 3.36%, quinones 0.56%, sphingolipids 0.08%, steroids 0.26%, tannins 1.12%, terpenoids 3.04%, and tryptamines 0.16%. Isoflavones, brazilin, and 3,7-di-O-methylquercetin were not detected in the Japan Shizuoka matcha (R). Delphinidin 3-sambubioside was not detected in the China Tongren matcha (G). Kaempferol 3-O-beta-D-glucopyranosyl-7-O-alpha-L-rhamnopyranoside, 3,7-di-O-methylquercetin, and glycitin were not detected in the China Wuyi matcha (Z2).

Kaempferol-7-O-rhamnoside, 3,7-Di-O-methylquercetin, epigallocatechin gallate, epicatechin gallate, and epigallocatechin were detected in the China Hangzhou matcha (Z1), China Wuyi matcha (Z2), Japan Shizuoka matcha (R), China Enshi matcha (H), and China Tongren matcha (G), respectively. The 2D structures of kaempferol-7-O-rhamnoside, 3,7-Di-O-methylquercetin, epigallocatechin gallate, epicatechin gallate, and epigallocatechin can be seen in Appendix A.

### 3.3. PCA of Non-Volatile Substances of Matcha from Different Geographic Regions

Total ion chromatography (TIC) is a spectrum obtained by the continuous summation of the strength of all ions in a mass spectrometer at each time point, which includes the ion strength and the retention time (RT) of metabolites of each component in the chromatograph [1,9,13]. As can be seen in Figure 3a, the TIC diagram results showed that the data recorded in this experiment have good repeatability and reliability. This indicated that the non-volatile substance detection data of the matcha samples from different geographic regions are reliable.

Principal component analysis (PCA) is a very classical dimensionality reduction statistical method for extracting important information. Through orthogonal transformation, a group of possibly correlated variables was transformed into a group of linearly uncorrelated variables with PCA [20]. As shown in Figure 3b, the non-volatile substance composition and content of R and Z1, Z2, G, and H were significantly distinguished. Group 1 included the China Hangzhou matcha (Z1). Group 2 included the China Wuyi matcha (Z2) accession with Japan Shizuoka matcha (R). Group 3 included the China Enshi matcha (H). Group 4 included the China Tongren matcha (G). These four groups could be easily distinguished from each other. In group 2, the China Wuyi matcha (Z2) overlapped with the Japan Shizuoka matcha (R), indicating that their flavor substance profiles were similar. The cumulative variance contribution of the first principal components was 35.7% (PC1). The cumulative variance contribution of the second principal components was 22.57% (PC2).

### 3.4. Metabolic Profile Analysis of Differential Non-Volatile Substances of Matcha (Z1 vs. Z2, R as the Control)

Hangzhou matcha (Z1) and Wuyi matcha (Z2) are representative matcha from eastern China. The characteristics of the non-volatile metabolites of these matcha samples were analyzed with reference to the Japan Shizuoka matcha (R). The differences in metabolite species and relative contents among Z1, Z2, and R were compared using clustering heat map analysis methods. As shown in Figure 4a, a total of 1243 metabolites differed among the three types of matcha (Z1 vs. Z2 vs. R). The metabolites were 352 amino acids and derivatives, 185 organic acids, 116 benzenes and substituted derivatives, 95 flavonoids, 52 alkaloids, 46 phenolic acids, 32 terpenoids, 30 heterocyclic compounds, 30 lipids, 30 nucleotides and derivatives, 25 alcohols and amines, 25 GPs, 18 lignans and coumarins, 10 FAs, six GLs, six quinones, six steroids, six tannins, one tryptamine, and 163 others (Figure 4a). The non-volatile metabolites for which Z2 was higher than R were 123 amino acids and derivatives, 102 organic acids, 66 benzenes and substituted derivatives, 63 flavonoids, 24 alkaloids, 23 phenolic acids, 11 terpenoids, 15 heterocyclic compounds, 13 lipids, 12 nucleotides and derivatives, 16 alcohols and amines, 13 GPs, five lignans and coumarins, three FAs, three GLs, two quinones, two steroids, two tannins, and 101 others. The non-volatile metabolites for which Z1 was higher than R were 68 amino acids and derivatives, 56 organic acids, 47 benzenes and substituted derivatives, 42 flavonoids, 13 alkaloids, 11 phenolic acids, four terpenoids, seven heterocyclic compounds, four lipids, six nucleotides and derivatives, four alcohols and amines, 13 GPs, two lignans and coumarins, one fatty acid (FA), one glycerol (GL), one quinone, two steroids, two tannins, and 35 others. The non-volatile metabolites for which Z2 was higher than Z1 were 83 amino acids and derivatives, 46 organic acids, 26 benzenes and substituted derivatives, 32 flavonoids, 14 alkaloids, 10 phenolic acids, three terpenoids, four heterocyclic compounds, two lipids, two nucleotides and derivatives, two alcohols and amines, one quinone, two steroids, two tannins, and 25 others. There were differences in the type and content of non-volatile metabolites between Z1 and Z2 compared to R. Overall, the content of non-volatile difference metabolites for Z2 was higher than that for Z1.

The correlation coefficient between the non-volatile metabolites of Z1 and Z2 was calculated using correlation analysis, which showed the internal structure of the data with a network diagram (Figure 4b). The interactions between the relevant feature nodes were revealed by using elements, such as a point and connecting line color, size, and thickness. The relationship between the non-volatile metabolites of Z1 and Z2 is shown in Figure 4b. The key interacting metabolites between Z1 and Z2 are Val-Leu-Leu-Val-Val, acetylvalerenolic acid, Val-Leu-Leu-Val-Val, acetylvalerenolic acid, 9-α,10-β-52-[(1R,2R)-3-oxo-2-[(Z)-5-[3,4,5-trihydroxy-6-(hydroxymethyl)oxan-2-yl]oxypent-2-enyl]cyclopentyl]acetic acid, Glu-Asn-Ile-Ile-Asp helicide, 5,6,7-trihydroxy-2-(4-hydroxyphenyl)-8-(3,4,5-trihydroxyoxan)-4H-chromen-4-one, Ile-His, docosa-2,4,6,8,10,12-hexaenoic acid, D-mannitol 1-phosphate, D-mannitol 1-phosphate, 7-5-hydroxy-2-(4-hydroxyphenyl)-3-{[3,4,5-trihydroxy-6-(hydroxymethyl)oxan-2-yl]oxy}-4H-chromen-4-one, 3-(3,4-dimethoxyphenyl)-N-[2-(3,4-dimethoxyphenyl)-2-ethyl]prop-2-enimidic acid, 3,4,5-trihydroxybenzoate, 3-[4-(2-hydroxyethyl)piperazin-1-ium-1-yl]propane-1-sulfonate, 6-(4-{5,7-dihydroxy-4-oxo-6,8-bis[3,4,5-trihydroxy-6-(hydroxymethyl)oxan-2-yl]-3,4-dihydro-2H-1-benzopyran-2-yl}phenoxy)-3,4,5-trihydroxyoxane-2-carboxylic acid, cycloplazonic acid, His-Val-Gln-Arg, geranyl acetate, n-acetyl-L-arginine, n-acetyl-L-phenylalanine, didymin, Pro-Gly-Asn, Ile-Asn-Phe, nordihydrocapsiate, 1-(4-hydroxy-3-methoxyphenyl)-3-decanone, bracteatin, ostruthin, pantetheine-cinnamoyloxytaxa-4(20),11-diene-13alpha-ol, 2-[(1R,2R)-3-oxo-2-[(Z)-5-[3,4,5-trihydroxy--2-yl]oxypent-2-enyl]cyclopentyl]acetic acid, {[12-(6,11-dihydroxy-2,2,5-trimethyl-3,4,5,10-tetrahydro-4-yl)-6,11-dihydroxy-2,5-dimethyl-10-oxo}sulfonic acid, 7-oxan-2-yl]oxy}-5-hydroxy-2-(4-hydroxyphenyl)-3-4H-chromen-4-one, 5,6,7-trihydroxy-2-(4-hydroxyphenyl)-8-(3,4,5-trihydroxyoxan-2-yl)-4H-chromen-4-one, 3-(3,4-dimethoxyphenyl)-N-[2-(3,4-dimethoxyphenyl)-2-(sulfooxy)ethyl]prop-2-enimidic acid, and Glu-Asn-Ile-Ile-Asp helicide.

OPLS-DA, a supervised discriminant analysis method, was used to analyze the non-volatile differential metabolites of Z1 and Z2 (Figure 4c). The relationship between non-volatile metabolites and matcha sample classes from different regions was discussed (Figure 4c,d). As can be seen from Figure 4c, Z1, Z2, and R were effectively separated. The T score and the orthogonal T score accounted for 27% and 10.2% of the total variance, respectively. The high predictability (Q^2^) and strong goodness of fit (R^2^ X, R^2^ Y) of the OPLS-DA models were observed in comparisons among Z1, Z2, and R (Q^2^ = 0.907, R^2^ X = 0.371, R^2^ Y = 1.0).

To gain an in-depth understanding of the metabolite differences among Z1, Z2, and R, differentiable metabolite screening was performed, with all metabolites annotated in accordance with variable importance in projection (VIP) scores (Figure 4d). The criteria for significant differences included a VIP score of ≥1. There were 20 key differentiating non-volatile metabolites compounds with such a VIP score: docosa-2,4,6,8,10,12-hexaenoic acid, 3-[(2R,3R,4R,5S)-3,4-dihydroxy-5-(hydroxymethyl)oxolan-2-yl]oxy-2-(3,4-dihydroxyphenyl)-5,7-dihydroxychromen-4-one, cis-11-methyl-2-dodecenoic acid, Met-Ser-His, 5,6,7-trihydroxy-2-(4- hydroxyphenyl)-8-(3,4,5-trihydroxyoxan-2-yl)-4H-chromen-4-one, didymin, cycloplazonic acid, 6-(4-phenoxy)-3,4,5- trihydroxyoxane-2-carboxylic acid, Ile-His, helicide, neotame, entylacetic acid, acetylvalerenolic acid, Glu-Asn-Ile-Ile-Asp, 2-(ethylsulfonyl)ethanol, quercetin, [2,3,5-trihydroxy-6-(hydroxymethyl)oxan-4-yl]-3,4,5-trihydroxybenzoate, 5,7-dihydroxy-3-(7- hydroxy-5-methoxy-2,2-dimethyl-3,4-dihydro-2H-1-benzopyran-6-yl)-4H-chromen-4-onem, 2-[(1R,2R)-3-oxo-2-[(Z)-5-[3,4,5-trihydroxy-6-(hydroxymethyl)oxan-2-yl] oxypent-2-enyl]cyclop n-acetyl-L-arginine, and geranyl acetate.

KEGG pathway diagrams are often used to visually represent the upstream and downstream relationships of metabolites [33]. As shown in Figure 4e,f, the metabolic pathways, biosynthesis of secondary metabolites, biosynthesis of co-factors, flavonoid biosynthesis, biosynthesis of amino acids, biosynthesis of various plant secondary metabolites, and purine metabolism of metabolic pathways were the main KEGG pathways. The cluster frequencies of the metabolic pathways, biosynthesis of secondary metabolites, biosynthesis of co-factors, flavonoid biosynthesis, biosynthesis of amino acids, biosynthesis of various plant secondary metabolites, and purine metabolism were 72.03%, 50.00%, 14.41%, 9.32%, 8.47%, and 8.47%, respectively.

### 3.5. Metabolic Profile Analysis of Differential Non-Volatile Substances of Matcha (G vs. H, R as the Control)

Hubei Enshi and Guizhou Tongren are typical of matcha from central and western China. The characteristics of the non-volatile metabolites of the matcha from these regions were analyzed using Japan Shizuoka matcha (R) as the control using clustering heat map analysis methods. As shown in Figure 5a, a total of 1617 metabolites differed among the three types of matcha (G vs. H vs. R), comprising 464 amino acids and derivatives, 275 organic acids, 116 benzenes and substituted derivatives, 111 flavonoids, 62 alkaloids, 55 phenolic acids, 53 nucleotides and derivatives, 41 lipids, 39 terpenoids, 30 nucleotides and derivatives, 35 heterocyclic compounds, 34 glycerol phosphates (GPs), 29 alcohols and amines, 27 lignans and coumarins, 16 FAs, seven GLs, four quinones, seven steroids, nine tannins, one tryptamine, and 186 others (Figure 5a). The non-volatile metabolites for which G was higher than R were 298 amino acids and derivatives, 195 organic acids, 89 benzenes and substituted derivatives, 93 flavonoids, 46 alkaloids, 36 phenolic acids, 21 nucleotides and derivatives, 16 lipids, 17 terpenoids, 13 nucleotides and derivatives, 19 heterocyclic compounds, 11 GPs, 10 alcohols and amines, eight lignans and coumarins, four FAs, three GLs, two quinones, two steroids, four tannins, and 113 others. The non-volatile metabolites for which H was higher than R were 102 amino acids and derivatives, 34 organic acids, 37 benzenes and substituted derivatives, 26 flavonoids, 13 alkaloids, 13 phenolic acids, eight nucleotides and derivatives, five lipids, three terpenoids, four nucleotides and derivatives, five heterocyclic compounds, two GPs, one alcohol and amine, two lignans and coumarins, and 58 others. There were noticeable differences in the type and content of non-volatile metabolites between G and H compared to R. Overall, the content of non-volatile difference metabolites for G was higher than that for H.

The correlation coefficient between non-volatile metabolites of G and H was calculated using correlation analysis, which showed the internal structure of the data with a network diagram (Figure 5b). The interactions between the relevant feature nodes are shown by using elements such as a point and connecting line color, size, and thickness. The relationship between the non-volatile metabolites of G and H can be seen in Figure 5b. The key interaction metabolites (VIP score ≥ 1) between G and H were mulberrofuran C, glycyl-L-tyrosine, stearidonic acid ethyl ester, dTDP-3-oxo-2,6-dideoxy-D-glucose, corilagin, His-Val-Gln-Arg, Trp-Trp, Ile-Tyr-Val, isococculidine, isoleucylaspartic acid, iosorhoifolin, lycoflexine, Lys-Glu-Arg, medicagenic acid, Met-Ser-His, Tyr-Asn-Gln-Asp, 2-methylanthraquinone, 4-O-methylgallic acid, corilagin, Ser-Tyr-Lys-Asp, n-methyl-L-proline,2,2,5-trimethyl-10-oxo-3,4,5,10-tetrahydro-2H-1-oxa-5-azatetraphen-4-yl)-6,11-dihydroxy-2,5-dimethyl-10-oxo-5,10-dihydro-2H-1-oxa-5-azatetraphen-2-yl]methoxy}sulfonic acid, betulinic acid, cis-4-decen-1-ol, 3,4,5-trihydroxyoxane-2-carboxylic acid, 2,3-dihydro-1-benzofuran-2-yl)methyl]phenyl}oxidanesulfonic acid, oxidanesulfonic acid, 4,6-dihydroxy-2-(hydroxymethyl)-5-(3,4,5-trihydroxybenzoyloxy)oxan-3-yl,3,4,5-trihydroxybenzoate-6-3,4,5-trihydroxyoxane-2-carboxylic acid, 7-4H-chromen-4-one arcaine, sulfonic acid, 2-amino-acetic acid, and 6-oxy-3,4,5-trihydroxyoxane-2-carboxylic acid.

OPLS-DA was used to analyze non-volatile differential metabolites of G and H. The relationship between the non-volatile metabolites and matcha sample classes in different regions was discussed (Figure 5c). Z2 and R were effectively separated (Figure 5c, Z1). The T score and the orthogonal T score accounted for 38.1% and 22.2% of the total variance, respectively. High predictability (Q^2^) and strong goodness of fit (R^2^ X, R^2^ Y) of the OPLS-DA models were observed in comparisons among Z1, Z2, and R (Q^2^ = 0.941, R^2^ X = 0.603, R^2^ Y = 0.998).

To gain an in-depth understanding of the metabolite differences between G, H, and R, differentiable metabolite screening was performed, with all metabolites annotated in accordance with the variable importance in projection (VIP) scores (Figure 5d). The criteria for significant differences included a VIP score of ≥1. There were 20 key differentiating non-volatile metabolites compounds with such a VIP score: genistin, 6-3,4,5-trihydroxyoxane-2-carboxylic acid, corilagin, 6-oxy-3,4,5-trihydroxyoxane-2-carboxylic acid, N-methyl-L-proline, aine, Trp-Trp, Met-Ser-His, 7-5-hydroxy-2-(4-hydroxyphenyl)-3-4H-chromen-4-one, oxidanesulfonic acid, (3a,5b,7a,12a)-24-[(carboxymethyl)amino]-1,12-dihydroxy-24-oxocholan-3-yl-b-D-glucopyranosiduronic acid, lycoflexine, Thr-Glu-Leu-Lys, triethanolamine, Ile-Tyr-Val, and dihydroactinidiolide, dTDP-3-oxo-2,6-dideoxy-D-glucose.

KEGG pathway diagrams were employed to visually represent the upstream and downstream relationships of the non-volatile metabolites among G vs. H vs. R. As shown in Figure 5e,f, the metabolic pathways, biosynthesis of secondary metabolites, biosynthesis of co-factors, ABC transporters, nucleotide metabolism, biosynthesis of nucleotide sugars, glycerophospholipid metabolism, flavonoid biosynthesis, biosynthesis of amino acids, and purine metabolism were the main KEGG pathways. The cluster proportions of these pathways were 71.01%, 39.05%, 13.02%, 8.88%, 7.69%, 7.1%, 5.92%, 5.33%, and 5.33%.

## 4. Conclusions

In this study, the sensory quality characteristics and non-volatile substances of matcha processed with the same tea variety from different regions in China were found to be different. In the matcha samples, a total of 1131 distinct metabolites were identified, comprising 118 flavonoids, 14 tannins, 365 organic acids, 42 phenolic acids, 22 alkaloids, 39 saccharides, 208 amino acids and derivatives, 17 lignans and coumarins, seven quinones, 44 nucleotides and derivatives, 14 glycerophospholipids, two glycolipids, 15 alcohols and amines, 140 benzenes and substituted derivatives, 38 terpenoids, 30 heterocyclic compounds, and 15 lipids. Kaempferol-7-O-rhamnoside, 3,7-Di-O-methylquercetin, epigallocatechin gallate, epicatechin gallate, and epigallocatechin were the common metabolic substances detected in China Hangzhou matcha (Z1), China Wuyi matcha (Z2), China Enshi matcha (H), and China Tongren matcha (G). The content of non-volatile difference metabolites for G was higher than that for H. The 20 key differential non-volatile metabolites of Z1, Z2, G, and H were screened out separately, including docosa-2,4,6,8,10,12-hexaenoic acid, cis-11-methyl-2-dodecenoic acid, Met-Ser-His, 6-(4-phenoxy)-3,4,5- trihydroxyoxane-2-carboxylic acid, didymin, cycloplazonic acid, helicide, neotame, entylacetic acid, acetylvalerenolic acid, Glu-Asn-Ile-Ile-Asp, 2-(ethylsulfonyl)ethanol, and quercetin. The types of non-volatile flavor differential metabolites of G and H were more numerous than those of Z1 and Z2. The metabolic pathways, biosynthesis of secondary metabolites, biosynthesis of co-factors, flavonoid biosynthesis, biosynthesis of amino acids, biosynthesis of various plant secondary metabolites, and purine metabolism of metabolic pathways were the main KEGG pathways of different Chinese matcha.

## 5. Discussion

Matcha is one of the most popular tea products worldwide, being widely used in the food, beverage, health food, and cosmetic industries, among others [1,2,3,8,9,11,13]. Matcha maximizes the retention of tea polyphenols, caffeine, free amino acids, tea polysaccharides, chlorophyll, aromatic substances, fibroins, proteins, and a large number of minerals and vitamins and other bioactive substances, including water-soluble and insoluble substances, providing a full intake of tea leaf components. Previous studies have suggested that tea polyphenols, amino acids, flavonoids (including catechins), methylxanthines, phenolic acids, and caffeine are important non-volatile substances in matcha [8,9,13,41,44]. These non-volatile substances in the above-mentioned research accumulate within 30 kinds of matcha tea. In this study, an innovative non-targeted metabolomic approach was applied to the analysis of non-volatile substances in matcha tea. A number of previously unreported non-volatile substances were found in this study, such as organic acids, lignans and coumarins, and nucleotides and derivatives. The innovative findings of the discovery of a large number of non-volatile substances in matcha may provide a theoretical basis for basic research on the complex chemical substances in matcha.

The flavor quality of matcha is influenced by various factors such as the tea tree variety, planting area, agronomic practices, and processing techniques, which have been extensively studied [13,16,19,20]. Research on the effect of geography on matcha quality and the analysis of the mechanisms of variation is still relatively scarce. In one study, the chemical constituents of matcha collected from different Japanese regions were analyzed, which showed that the contents of polyphenols, amino acids, GABA, caffeine, catechins, phenolic acids, and other substances differed between regions [5,48]. In another study, the chemical composition and antioxidant activity of 12 matcha extract products from the Czech Republic market were analyzed, which showed significant differences in total phenols, flavonoids, flavonoids, phenolic acids, caffeine, etc. [43]. China is the largest producer of matcha and the largest consumer of matcha products in the world. However, differences in the chemical basis of matcha in different regions of China remain unclear. In this study, differences in the sensory quality and non-volatile flavor metabolites of Chinese matcha from four geographic regions were systematically investigated for the first time.

The different non-volatile metabolites associated with different geographic regions involve many different metabolic pathways. The metabolic pathways depend on each other to jointly maintain the flavor. This study provides new insights into the differences in metabolite profiles among typical Chinese matcha geographic regions with the same tea variety. This research aimed to provide a theoretical basis for the formation and control of the flavor quality of matcha and the development of new products. Matcha has become a very popular tea product in recent years and is one of the research hot spots in the field of tea science at home and abroad. This study focused on the flavor characteristics and the differences in non-volatile substances for four typical matcha production areas in China. The mechanisms of flavor difference and flavor formation in different typical matcha producing areas in China is still unclear. Further studies are still needed regarding the mechanisms of geographic influence on the flavor and non-volatile matter composition of matcha, as well as the mechanisms of geographic influence on the interaction between non-volatile matter and volatile matter.

## Figures and Tables

**Figure 1 foods-14-00097-f001:**
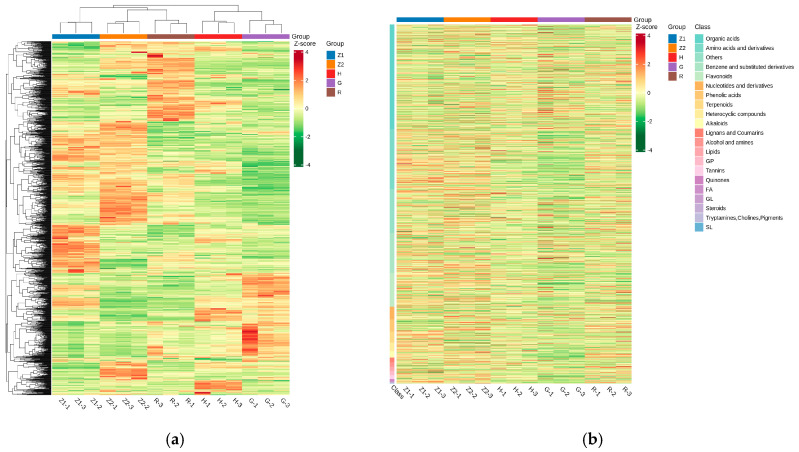
Heat map of non-volatile substances of matcha in different geographic regions. (**a**) is a heat map referring to the geographical cluster analysis of matcha samples, (**b**) is a heat map of the composition and content difference of the non-volatile substances in the matcha samples from different geographic regions. (Note: R refers to Japan matcha, G refers to China Tongren matcha, H refers to China Enshi matcha, Z1 refers to China Hangzhou matcha, Z2 refers to China Wuyi matcha).

**Figure 2 foods-14-00097-f002:**
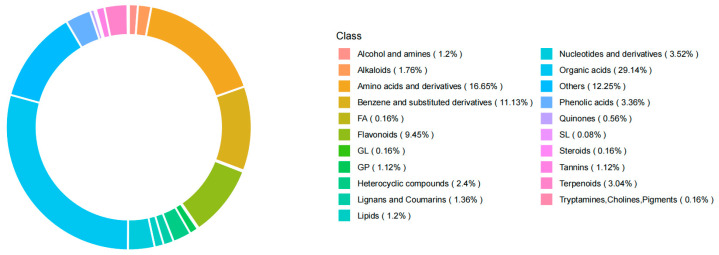
Proportional analysis of non-volatile substances in matcha from different geographic regions.

**Figure 3 foods-14-00097-f003:**
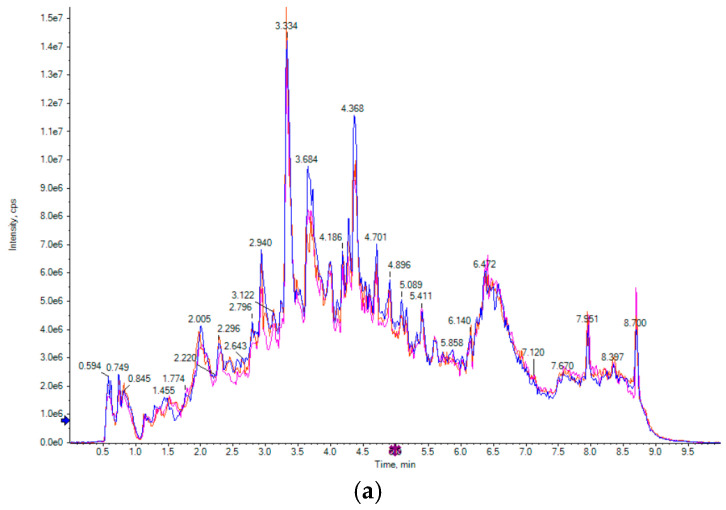
PCA analysis of non-volatile substances in matcha tea from different geographic regions. (Note: 1. R refers to Japan matcha, G refers to China Tongren matcha, H refers to China Enshi matcha, Z1 refers to China Hangzhou matcha, Z2 refers to China Wuyi matcha. 2. (**a**) Metabolite TIC multi-peak plot, (**b**) PCA analysis).

**Figure 4 foods-14-00097-f004:**
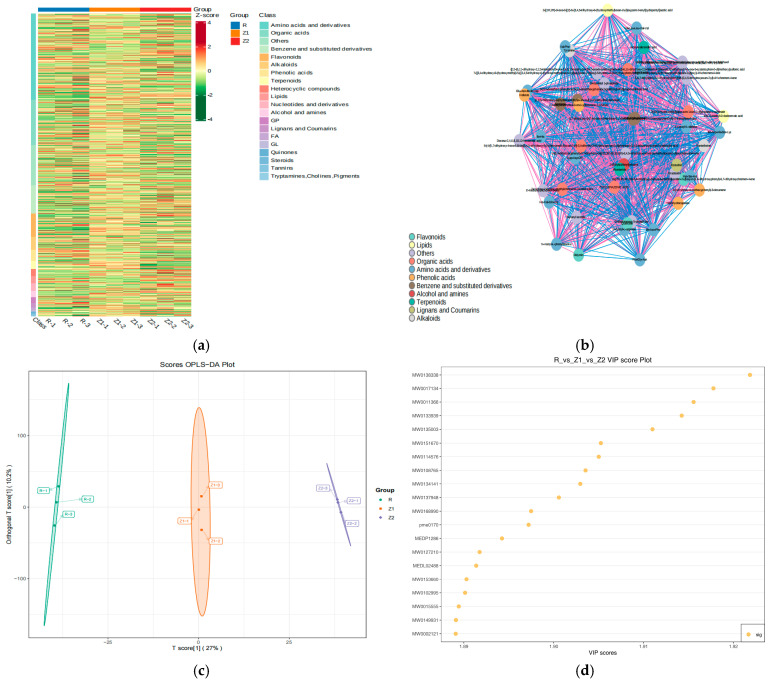
Analysis of differential non-volatile substances of matcha (R vs. Z1 vs. _Z2). (Note: (**a**) is all metabolites differed from R, Z1 and Z2. (**b**) is the internal structure of the data with a network diagram of R vs. Z1 vs. _Z2. (**c**) is the scores OPLS-DA plot of non-volatile metabolites of R vs. Z1 vs. _Z2. (**d**) is the VIP score plot of R vs. Z1 vs. _Z2. (**e**) is KEGG classification of R vs. Z1 vs. _Z2. (**f**) the *p*-value of main metabolic pathways of R vs. Z1 vs. _Z2. R refers to Japan matcha, Z1 refers to China Hangzhou matcha, Z2 refers to China Wuyi matcha).

**Figure 5 foods-14-00097-f005:**
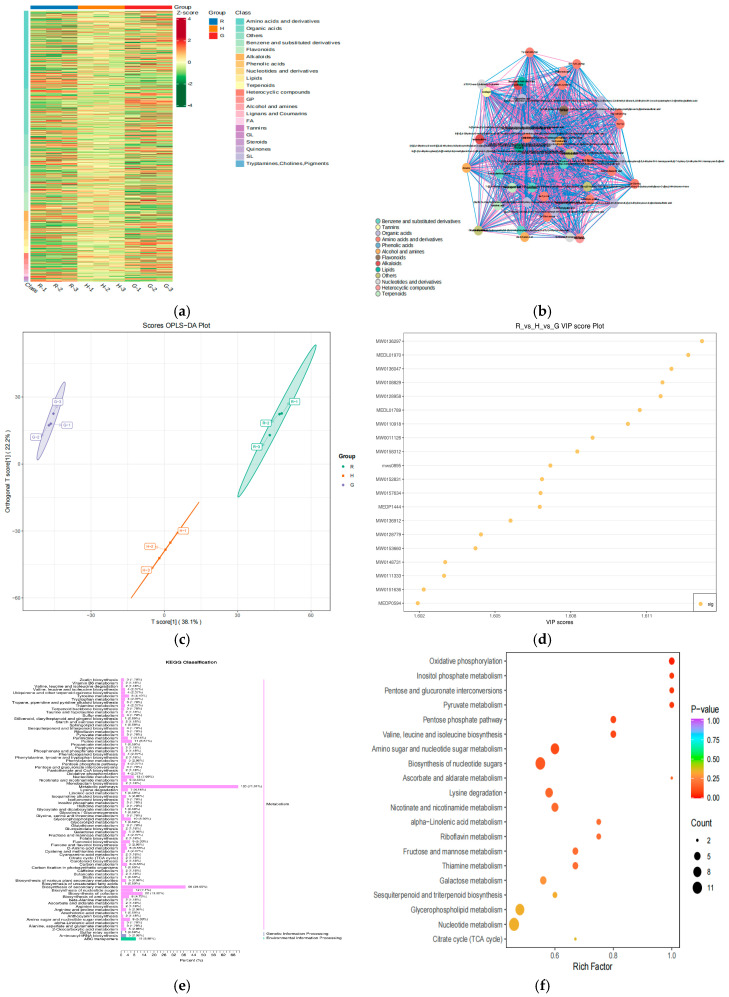
Analysis of differential non-volatile substances of matcha (R vs. H vs. G). (Note: (**a**) is all metabolites differed among R, H, G. (**b**) is the internal structure of the data with a network diagram of R vs. H vs. G. (**c**) is the scores OPLS-DA plot of non-volatile metabolites of R vs. H vs. G. (**d**) is the VIP score plot of R vs. H vs. G. (**e**) is KEGG classification of R vs. H vs. G. (**f**) the P-value of main metabolic pathways of R vs. H vs. G. R refers to Japan matcha, G refers to China Tongren matcha, H refers to China Enshi matcha).

**Table 1 foods-14-00097-t001:** Sensory quality of different geographic regions of matcha samples.

Different Geographically Regional Matcha	Appearance (10%)		Liquor Color (20%)		Aroma (35%)		Taste (35%)		Total Quality Score
	Comment	Score	Comment	Score	Comment	Score	Comment	Score	
Japan Shizuoka matcha (R)	delicate, even, vibrant green	93.8 ± 0.6 a	more intense green	94.3 ± 0.1 a	fresh, strong seaweed fragrance	93.6 ± 0.7 a	thick, mellow, fresh	94.1 ± 0.4 a	93.5 ± 0.7 a
China Hangzhou matcha (Z1)	delicate, even, green	91.3 ± 0.5 b	green	91.1 ± 0.8 c	slight seaweed fragrance, slightly smoky	85.2 ± 0.4 e	thick, coarse, bitter, astringent	83.5 ± 0.3 c	86.2 ± 0.4 c
China Wuyi matcha (Z2)	delicate, even, green	91.1 ± 0.2 b	intense green	92.5 ± 0.5 b	slight seaweed fragrance	88.3 ± 0.2 c	thick, fresh, slightly bitter	89.5 ± 0.6 b	89.1 ± 0.2 b
China Enshi matcha (H)	delicate, even, tender green	90.4 ± 0.3 b	intense green	92.8 ± 0.2 b	fresh, seaweed fragrance	91.5 ± 0.3 b	thick, fresh, slightly bitter	90.2 ± 0.5 b	90.7 ± 0.5 b
China Tongren matcha (G)	delicate, even, relatively tender green	88.6 ± 0.4 c	green	91.2 ± 0.6 c	slight seaweed fragrance	86.8 ± 0.5 d	thick, bitter, slightly astringent	84.5 ± 0.7 c	86.5 ± 0.6 c

Note: Different lowercase letters indicate significant differences in similar treatments (*p* < 0.01).

## Data Availability

The original contributions presented in this study are included in the article/Appendix A. Further inquiries can be directed to the corresponding authors.

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
