# Peer review of "Effect of Geographic Regions on the Flavor Quality and Non-Volatile Compounds of Chinese Matcha"

_foods, 2025, doi:10.3390/foods14010097_

Round 1
Reviewer 1 Report
Comments and Suggestions for Authors
The paper "Effect of geographic regions on the flavor quality and non-volatile compounds of China matcha" presents very extensive amount of data concerning matcha tea with different geographical origin. However, in my opinion, the manuscript has some obvious deficiencies. First of all, it is poorly written and sometimes very hard to read and understand. The English language and grammar should be thoroughly checked.
Some other comments are listed below:
-The phrase "tea leaves are picked and killed" and "steam killing" is used in the Introduction. Please explain.
-The Sample preparation part should be completely rewritten
-Lines 201-205- repetition of the same information
-Lines 205-216 - use of same sentences, hard to read and follow
-Lines 217-234 -the data should be presented more clearer, maybe with Table
-Lines 322-347 -unnecessary, maybe incorporate as supplementary material
Overall, the impression is that the Results section is not clearly written an that the paper needs detailed revision. Also, authors should incorporate Discussion as separate section in the manuscript.
Comments on the Quality of English Language
The English language and grammar should be thoroughly checked.
Author Response
Comments 1:The paper "Effect of geographic regions on the flavor quality and non-volatile compounds of China matcha" presents very extensive amount of data concerning matcha tea with different geographical origin. However, in my opinion, the manuscript has some obvious deficiencies. First of all, it is poorly written and sometimes very hard to read and understand. The English language and grammar should be thoroughly checked.
Response 1:Thank you very much for your good suggestions. According to your suggestion, we have polished our manuscript carefully and corrected the grammatical, styling, and typos found in the manuscript. We try our best to make it easy to read and understand. The English language and grammar have been thoroughly checked. Red font parts were the result of modified.
Comments 2:-The phrase "tea leaves are picked and killed" and "steam killing" is used in the Introduction. Please explain.
Response 2:Thank you very much for your constructive suggestions. According to your suggestion, the mistakes have been revised. "tea leaves are picked and killed" and "steam killing" should be revised for “steaming fixation”. Red font parts were the result of modified.
Comments 3:-The Sample preparation part should be completely rewritten
Response 3:Thank you very much for your constructive suggestions. According to your suggestion, the sample preparation part have been completely rewritten. Red font parts are the result of modified.
Comments 4:-Lines 201-205- repetition of the same information
Response 4:Thank you very much for your constructive suggestions. Due to the complexity of sensory evaluation terms, we have accepted your suggestion and put these sensory qualitative terms in Table 1. Red font parts are the result of modified.
Comments 5:-Lines 205-216 - use of same sentences, hard to read and follow
Response 5:Thank you very much for your constructive suggestions. Due to the complexity of sensory evaluation terms, we have accepted your suggestion and put these sensory qualitative terms in Table 1. Red font parts are the result of modified.
Comments 6:-Lines 217-234 -the data should be presented more clearer, maybe with Table
Response 6:Thank you very much for your constructive suggestions. Due to the complexity of sensory evaluation terms, we have accepted your suggestion and put these sensory qualitative terms in Table 1.
Comments 7:-Lines 322-347 -unnecessary, maybe incorporate as supplementary material
Response 7:Thank you very much for your constructive suggestions. Normal have been revised to normal. Red font parts are the result of modified.
Comments 8:Overall, the impression is that the Results section is not clearly written an that the paper needs detailed revision. Also, authors should incorporate Discussion as separate section in the manuscript.
Response 8:Thank you very much for your constructive suggestions. According to the suggestion, the results section have been revised in detail and comprehensively. Based on your comments and the format requirements of the Foods journal, we put the discussion session content in the conclusion module. Hope to get your understanding and recognition. Red font parts are the result of modified.

Reviewer 2 Report
Comments and Suggestions for Authors
Review of the manuscript “Effect of geographic regions on the flavor quality and non-volatile compounds of China matcha” by Hongchun Cui, Yun Zhao, Hongli Li, Min Ye, Jizhong Yu and Jianyong Zhang . Jianyong Zhang submitted to the Foods journal. foods-3336403
This study evaluates matcha's sensorial quality and metabolite profile of leaves obtained from different geographic regions by liquid chromatography-mass spectrum and PC analysis. The topic of the research is interesting.
The work is well-founded and written, but I suggest a revision of the English language to avoid errors. English language should be checked; some sentences are unclear.
Reinforce why Japanese matcha (R) was used as a control. I suggest mentioning the methodology of PC analysis.
The author indicates several metabolites in the “3.2 cluster analysis… section” to determine differences in matcha from different geographical regions. However, the proportions of metabolites in R (control) are not indicated. On the other hand, it would be advisable to include the 2D or 3D structures of metabolites 1122, 1111, 1114, 1125, and 1116.
Please standardize the font.
Reinforce the description of Figure 3a.
PC1 describes the most significant features of the original data matrix (275) and explains 31.63% of the characterization of non-volatile metabolites (284-285), and so on. Recheck.
Improve the wording of sections 3.4 and 3.5. Enlarge the images in Figures 4 and 5.
Improve the description of each image of the figures.

Author Response
Comments 1: The work is well-founded and written, but I suggest a revision of the English language to avoid errors. English language should be checked; some sentences are unclear.
Response 1: Thank you very much for your good suggestions. According to the suggestion, we have polished our manuscript carefully and corrected the grammatical, styling, and typos found in the manuscript. The unclear sentences have been revised. Red font parts are the result of modified.
Comments 2: Reinforce why Japanese matcha (R) was used as a control. I suggest mentioning the methodology of PC analysis.
Response 2: Thank you very much for your constructive suggestions. China is the world's largest producer of matcha and consumer of matcha products. The production of Japan matcha is second only to China. Considering the differences in production, production process, flavor and application areas between Chinese matcha and Japanese matcha, Japanese matcha was selected as a control. The differences in flavor characteristics of matcha products from different geographic regions in China were systematically analyzed. And, the differences in flavor characteristics with Japanese matcha were also be studied. We hope to provide theoretical bases for the research on the chemistry of flavor quality and the development of new products of Chinese matcha. These sentence has been added to the manuscript.
Comments 3: The author indicates several metabolites in the “3.2 cluster analysis… section” to determine differences in matcha from different geographical regions. However, the proportions of metabolites in R (control) are not indicated. On the other hand, it would be advisable to include the 2D or 3D structures of metabolites 1122, 1111, 1114, 1125, and 1116.
Response 3: Thank you very much for your constructive suggestions. Figure 2 showed the overall proportion of non-volatile substances in H, G, Z1, Z2 and R, which have included the proportion of non-volatile substances in R. The 2D structures of metabolites 1122, 1111, 1114, 1125, and 1116 have been added. The 2D structure of 1122, 1111, 1114, 1125, and 1116 could been seen in figure S1.
Comments 4: Please standardize the font.
Response 4: Thank you very much for your good suggestions. According to your suggestion, the fonts have been revised in detail.
Comments 5: Reinforce the description of Figure 3a.
Response 5: Thank you very much for your constructive suggestions. According to your suggestion, the description of Figure 3a have been explained and reinforced.
Comments 6: PC1 describes the most significant features of the original data matrix (275) and explains 31.63% of the characterization of non-volatile metabolites (284-285), and so on. Recheck.
Response 6: Thank you very much for your constructive suggestions. The line 275, 284-285 have been revised.
Comments 7: Improve the wording of sections 3.4 and 3.5. Enlarge the images in Figures 4 and 5.
Response 7: Thank you very much for your constructive suggestions. According to your suggestion, the wording of sections 3.4 and 3.5 have been improved. The images in Figures 4 and 5 have been enlarged.
Comments 8: Improve the description of each image of the figures.
Response 8: Thank you very much for your constructive suggestions. According to the suggestion, the description of each image of the figures have been improved.

Round 2
Reviewer 1 Report
Comments and Suggestions for Authors
Authors have incorporated some of the recommended suggestions. However, some responses remain unclear. Some of them are listed below:
Authors wrote: Response 7:Thank you very much for your constructive suggestions. Normal have been revised to normal. Red font parts are the result of modified.
Authors did not perform the revision according to my suggestions. They inserted a Figure of structures, while the text remained in the manuscript. Also, please explain the sentence "Normal have been revised to normal".
Authors wrote: Response 8:Thhank you very much for your constructive suggestions. According to the suggestion, the results section have been revised in detail and comprehensively. Based on your comments and the format requirements of the Foods journal, we put the discussion session content in the conclusion module. Hope to get your understanding and recognition. Red font parts are the result of modified.
The conclusion part of the manuscript is identical in the first and the second version. Also, according to journal rules, Discussion can be combined with Results, not the Conclusion section. As defined by the Journal Foods " Authors should discuss the results and how they can be interpreted in perspective of previous studies and of the working hypotheses. The findings and their implications should be discussed in the broadest context possible and limitations of the work highlighted." I believe that part is still missing.
Comments on the Quality of English LanguageI believe that the paper needs more professional English language editing. There are also typo, spelling and grammar errors in the manuscript.
Author Response
Comment 1:
Authors wrote: Response 7:Thank you very much for your constructive suggestions. Normal have been revised to normal. Red font parts are the result of modified.
Authors did not perform the revision according to my suggestions. They inserted a Figure of structures, while the text remained in the manuscript. Also, please explain the sentence "Normal have been revised to normal".
Reply 1:
Thank you very much for your good suggestion. Firstly, the ‘metabolites 1122, 1111, 1114, 1125, and 1116’ have be replaced by ‘kaempferol-7-O-rhamnoside, 3,7-Di-O-methylquercetin, epigallocatechin gallate, epicatechin gallate, epigallocatechin’ in the line 27-28 and 242-243 of the new edition manuscript. The 2D structure of 1122, 1111, 1114, 1125, and 1116 could been seen in figure S1. Secondly, ‘Normal have been revised to normal’ was a clerical error. I am sorry for the inconvenience.
Comment 2:
Authors wrote: Response 8:Thhank you very much for your constructive suggestions. According to the suggestion, the results section have been revised in detail and comprehensively. Based on your comments and the format requirements of the Foods journal, we put the discussion session content in the conclusion module. Hope to get your understanding and recognition. Red font parts are the result of modified.
The conclusion part of the manuscript is identical in the first and the second version. Also, according to journal rules, Discussion can be combined with Results, not the Conclusion section. As defined by the Journal Foods " Authors should discuss the results and how they can be interpreted in perspective of previous studies and of the working hypotheses. The findings and their implications should be discussed in the broadest context possible and limitations of the work highlighted." I believe that part is still missing.
Reply 2:
Thank you very much for your constructive suggestions. According to your comments, The results and how they can be interpreted in perspective of previous studies and of the working hypotheses have been discussed. The broadest context possible and limitations of the work highlighted have also been discussed. In addition, we made further revisions and refinements to the English writing of the manuscript.

Reviewer 2 Report
Comments and Suggestions for Authors
The authors' responses have clarified several points raised in the previous review. Modifications in the document are identified, and the authors implement them in the new text version. The document has been improved compared to the revised version.
Author Response
Comment: The authors' responses have clarified several points raised in the previous review. Modifications in the document are identified, and the authors implement them in the new text version. The document has been improved compared to the revised version.
Reply: Thank you very much for your approval.
Round 3
Reviewer 1 Report
Comments and Suggestions for Authors
Conclusions and discussion should not be merged as a one subtitle. Discussion should be before the conclusion section, as separate subtitle.
Author Response
Comment 1:
Authors wrote: Conclusions and discussion should not be merged as a one subtitle. Discussion should be before the conclusion section, as separate subtitle.
Reply 1:
Thank you very much for your good suggestion. Conclusions and discussion have been merged as a one subtitle. Discussion have been before the conclusion section, as separate subtitle. Conclusions and discussion have been revised again. In addition, the English writing of the entire manuscript has also been requested by the MDPI Author Service Center.